# Preparation and Biomedical Applications of Cucurbit[n]uril-Based Supramolecular Hydrogels

**DOI:** 10.3390/molecules28083566

**Published:** 2023-04-19

**Authors:** Ruihan Gao, Qingmei Ge, Hang Cong, Yunqian Zhang, Jianglin Zhao

**Affiliations:** 1Enterprise Technology Center of Guizhou Province, Guizhou University, Guiyang 550025, China; 2Key Laboratory of Macrocyclic and Supramolecular Chemistry of Guizhou Province, Guizhou University, Guiyang 550025, China; 3Precision Medicine R&D Center, Zhuhai Institute of Advanced Technology, Chinese Academy of Sciences, Zhuhai 519000, China

**Keywords:** cucurbit[n]urils, supramolecular hydrogels, biomedical, host–guest interactions

## Abstract

The cucurbit[n]uril supramolecular hydrogels are driven by weak intermolecular interactions, of which exhibit good stimuli responsiveness and excellent self-healing properties. According to the composition of the gelling factor, supramolecular hydrogels comprise Q[n]-cross-linked small molecules and Q[n]-cross-linked polymers. According to different driving forces, hydrogels are driven by the outer-surface interaction, the host–guest inclusion interaction, and the host–guest exclusion interaction. Host–guest interactions are widely used in the construction of self-healing hydrogels, which can spontaneously recover after being damaged, thereby prolonging their service life. The smart Q[n]s-based supramolecular hydrogel composed is a kind of adjustable and low-toxicity soft material. By designing the structure of the hydrogel or modifying the fluorescent properties, etc., it can be widely used in biomedicine. In this review, we mainly focus on the preparation of Q[n]-based hydrogels and their biomedical applications including cell encapsulation for biocatalysis, biosensors for high sensitivity, 3D printing for potential tissue engineering, drug release for sustained delivery, and interfacial adhesion for self-healing materials. In addition, we also presented the current challenges and prospects in this field.

## 1. Introduction

Self-assembly is ubiquitous. Molecular self-assembly mainly diffuses molecules to form nanostructures through a series of non-covalent interactions, such as hydrogen bonding, the hydrophobic interaction, π-π stacking interaction, electrostatic interaction, and van der Waals forces [1,2]. It can be used to construct a wide variety of nanostructures, including the simplest one-dimensional structures, which can be spontaneously converted into thermodynamically stable two-dimensional and three-dimensional structures, including ordered fibrous hydrogels [3,4]. Compared with traditional polymer gels, supramolecular hydrogels exhibit good reversibility and stimuli responses. Reversible supramolecular hydrogels are easy to handle, have good self-recovery, excellent self-repair, good stimulus responses, and controlled injections [5].

Cucurbit[n]uril (Q[n]) is a macrocyclic compound constructed by methylene-bridged urea units [6,7]. It displays a highly symmetrical rigid structure with a pumpkin-like shape, an inner hydrophobic cavity, and hydrophilic carbonyl groups distributed on both sides of its port. The surface electrostatic potential calculation results show that the inner surface of the cucurbit[n]uril cavity is nearly electrically neutral; the port is negatively charged, owing to the carbonyl oxygen; and the outer surface is positively charged [8,9,10]. Regarding its structure and surface electrostatic potential characteristics, the cavity of cucurbit[n]uril contains guest molecules, the port carbonyl coordinates with the metal ion/protonated guest, and the outer-surface interactions coordinate with the anion or negatively charged substance. Consequently, Q[n]s complexation can provide reversible crosslinking which can be manipulated by a competitive guest as a stimulus [11,12] to endow a hydrogel network with desirable dynamic properties [13]. These also are the main driving forces used for the formation of Q[n]-based supramolecular hydrogels [14,15,16,17,18]. We believe that Q[n] is an ideal basic building block for various frame structures. Therefore, compared with other host molecules, cucurbit[n]urils feature more modes of action and can exhibit a wider range of field applications. The object recognition property of Q[n]s is closely related to their cavity size. Presently, Q[n]-based supramolecular hydrogels comprise Q[5], Q[6], Q[7], and Q[8]. Though the change in the structure of guest molecules (coordination atoms, numbers) and external environmental factors (pH, illumination, temperature, ionic strength, ultrasound, enzyme, and solvent polarity), the self-assembly process and nanostructure of cucurbit[n]uril can be regulated, thereby broadening the field applications of cucurbit[n]uril hydrogel materials [19].

In this paper, the construction and application of Q[n]-based supramolecular hydrogels are widely summarized. According to the composition of the gelator, Q[n]-based supramolecular hydrogels can be formed through the self-assembly of a low molecular gelator and via the non-covalent crosslinking of polymer chains modified with functional groups. The research on hydrogels constructed by small molecules mainly focused on the selection and design of gelators, the process of self-assembly, and the diversity of structures. The research on supramolecular hydrogels constructed by polymers mainly focused on the construction method of supramolecular networks, the performance of gels and their practical application values. The role of Q[n]s in supramolecular hydrogels provides a theoretical and practical basis for subsequent research and practical applications.

## 2. Q[n]-Cross-Linked Polymer-Based Supramolecular Hydrogels

Q[n]-cross-linked polymer-based supramolecular hydrogels are the main building blocks of Q[n]-based hydrogels. Polymers can be constructed from natural macromolecules and synthetic monomers. Natural polymers exhibit good biocompatibility and biodegradability and can be widely used in the biological field. The monomers of synthetic polymers can be designed and customized based on demand. Q[n]-cross-linked polymer hydrogels are based on non-covalent crosslinks between cucurbit[n]uril and guest molecules. Due to the physical properties of non-covalently cross-linked hydrogels, most prepared hydrogel materials exhibit excellent responsiveness, self-healing ability, and have broad field prospects in the field of biomaterial science [20]. Unlike macrocyclic compounds such as crown ethers, cyclodextrins, and calixarenes, Q[n] features a hydrophobic cavity and a strong polar port structure. The host–guest complex formed by Q[n]s and organic guest molecules, particularly those with amine groups, can undergo protonation and exhibit unique properties. These guest molecules mainly include amantadine [21], bipyridine [22], viologen derivatives [23], etc. According to different physical crosslinking methods, the cross-linked hydrogels of Q[n]s and polymers are driven by the polymer host–guest exclusion interaction and polymer host–guest inclusion interaction.

### 2.1. Hydrogels Driven by the Polymer Host–Guest Exclusion Interaction

Li et al. prepared a novel supramolecular gel via a hydrogen-bonding interaction between the carbonyls of Q[7] and the amino groups of polyacrylamide (PAAm). As shown in Figure 1, a hydrogel formed through the crosslinking of polymer chains with Q[7]. The amino group of PAAm is hydrogen bonded to the port carbonyl of Q[7] through an exclusion interaction, which is the main driving force for supramolecular hydrogel formation. The Q-PAAm gel exhibited excellent mechanical properties. The pH could control the sol–gel transition and self-assembly of the Q-PAAm gel [24].

### 2.2. Hydrogels Driven by the Polymer Host–Guest Inclusion Interaction

Supramolecular hydrogels are generally formed through the host–guest inclusion between Q[n]s and the side chain groups of polymer chains [25,26,27,28]. The dynamic Q[n] host–guest interactions could effectively achieve energy dissipation through the reversible association/dissociation of the host–guest complexes, thus contributing to the toughness of these supramolecular hydrogel networks.

Scherman et al. reported several studies on the preparation of supramolecular hydrogels through the host–guest interactions between Q[8] and specific guest groups [28,29,30,31,32,33,34]. In 2010, a self-healing hydrogel based on the 1:1:1 supramolecular ternary complexes of Q[8]/viologen/naphthoxy was reported [29]. Then, they constructed a double-network supramolecular hydrogel through the hydrogen bond interaction at the cohesive base-end of DNA and via the host–guest interaction between Q[8] and phenylalanine-functionalized carboxymethyl cellulose (CMC-phe), as shown in Figure 2 [33]. The double-network hydrogel exhibited enhanced strength and thermal stability, good stretchability, ductility, shear thinning, and thixotropy. In addition, nuclease and cellulase could degrade the two hydrogel networks; therefore, the performance of the hydrogel system is precisely regulated. Hydrogel can be used as a soft scaffold material. Similarly, they also developed a microfluidic platform for the continuous fabrication of double-network hydrogel microfibers with tunable structural, chemical, and mechanical features [28]. Construction of the double-network microfibers was achieved through the incorporation of dynamic Q[n] host–guest interactions, as energy-dissipation moieties, within an agar-based brittle network. These microfiber networks exhibited high fracture stress, stretchability, and toughness by two-to-three orders of magnitude compared to the pristine agar network, while simultaneously gaining recoverable hysteretic energy dissipation without losing their mechanical strength. This study provides a versatile method for the continuous fabrication of soft materials with regulated functions.

Q[8] can simultaneously bind with two guests in its portal to form a ternary complex. Webber et al. [35] proposed a method to photoswitch between the physical and chemical crosslinking of bright orange hydrogels from a 4-arm PEG macromer appended with Brooker’s Merocyanine, using Q[8] as a template to form physical crosslinks and to drive photodimerization, converting the hydrogel crosslinking from physical to chemical networks, as shown in Figure 3. Additionally, they also studied the dynamics of the hydrogel network by controlling the external factors. Liu et al. [36] prepared a microsphere composite hydrogel through the dynamic host–guest interactions between 1-benzyl-3-vinylimidazolium bromide units bonded to chitosan microspheres and the host molecule Q[8]. As polyfunctional initiating and crosslinking centers, the addition of chitosan microspheres greatly improved the mechanical properties of the hydrogel. The dynamic host–guest interaction endowed the hydrogel with a higher self-healing property. Owing to their excellent mechanical strength and self-healing properties, the hydrogels exhibit potential applications as biomaterials used for developing flexibility, sensors, and electronics.

### 2.3. Q[n]-Derivative-Based Supramolecular Hydrogels by the Host–Guest Inclusion Interaction

The surface of Q[n] is not easy to modify, thus limiting their derivatization and application. However, many studies have studied the surface modification of Q[n]s and achieved some results. Kim and co-workers developed an allyloxylated Q[6] monomer, which was surface-activated to modify Q[6] at the end of the polymer chain. As shown in Figure 4, they prepared polymer cross-linked hydrogels for an in situ assembly between Q[6]-grafted or -conjugated hyaluronic acid (Q[6]-HA) with polyamine-conjugated HA (PA-HA) by the host–guest interaction between Q[6] and PA. Furthermore, tag-Q[6] was introduced to prepare biocompatible hydrogels, which can be used in bioimaging, biosensors, drug delivery systems, and tissue engineering [37,38,39].

## 3. Q[n]-Cross-Linked Small-Molecule-Based Supramolecular Hydrogels

Small molecule hydrogels have attracted great interest from numerous researchers due to their unique structural properties, designability, and potential biological applications [40,41,42]. Various functional small molecular fragments can be directly used as the gelator of supramolecular hydrogels, reducing molecular design steps and avoiding more complex organic synthesis processes. However, there are relatively few reports on the direct construction of small organic molecule hydrogels using Q[n] as gelators limited by solubility. Moreover, the supramolecular hydrogel has poor mechanical properties and cannot withstand stretching and tearing tests, which greatly limits its application.

### 3.1. Hydrogels Driven by the Outer-Surface Interaction

The Q[n] outer surface exhibits a positive electrostatic potential and displays electrostatic interactions with other electronegative elements, including the ion–dipole interaction, dipole interaction, etc. Therefore, we propose the outer-surface interaction of Q[n]s. In 2007, Kim et al. [40] first reported that the hydrogel formed through the outer-surface interaction of Q[7]. By observing its crystal structure, it was found that Q[7] molecules formed a hydrogel through the outer-surface interaction. Q[7] molecules were stacked into one-dimensional supramolecular chains, cross-linked into a two-dimensional-layered structure, then cross-linked into a three-dimensional structure, as shown in Figure 5. The basic interaction between Q[7] can be summarized as follows: (1) the dipole interaction between the carbonyl oxygen at the Q[7] port with an electronegative electrostatic potential and the glycoside uridine methine/bridging methylene on the outer-space of the adjacent Q[7] with an electropositive electrostatic potential; (2) the hydrogen bond interaction between a large number of water molecules in the hydrogel system and the carbonyl oxygen at the end of Q[7]. Additionally, the results showed that Q[7]-based hydrogels exhibited excellent pH- and temperature-responsive properties.

### 3.2. Hydrogels Driven by the Simple Molecule Exclusion Interaction

Due to the different cavity sizes of Q[n]s, all guest molecules could easily enter Q[n]s with larger ports, including Q[7] and Q[8]. The study of Q[n]s-based supramolecular hydrogels showed that most hydrogels were prepared through the host–guest inclusion interaction. However, owing to their smaller sizes, Q[5] and Q[6], which are often excluded, have not received attention in the field of hydrogels. Recently, these hydrogels were constructed through the formation of a host–guest exclusion interaction based on the hydrogen bond and ion–dipole interaction between the guest and carbonyl oxygen atom of the Q[5] or Q[6] port.

In 2021, we found that a small molecular supramolecular hydrogel could be formed by the host–guest exclusion interaction between decamethylcucurbit[5]uril (Me_10_Q[5]) and *para*-phenylenediamine (*p*-PDA) [41]. Differently from the previously reported Q[n]-based supramolecular hydrogels, Q[n]s with small cavities could form one-dimensional molecular chains through the interaction with guest amino groups and transform into three-dimensional hydrogel materials. Therefore, competing species that can affect the exclusion interaction—such as metal ions, organic molecules, or even anions—will change the hydrogel morphology. Owing to its simple and low-cost synthesis method, Me_10_Q[5] has potential applications in the development of more practical and marketable Q[n]-based hydrogels.

In addition, as shown in Figure 6, we also prepared a low-molecular-weight supramolecular room-temperature phosphorescent hydrogel by hexmethyl cucurbit[5]uril (HmeQ[5]) and 1,4-diaminobenzene (DB) through a simple heating/mixing cooling method in HCOOH [42]. The results revealed that hydrogen bonding existed between the carbonyl group at the HmeQ[5] port and DB amino groups. The supramolecular gels were formed through a synergistic effect between the dipole–dipole interactions and outer-surface interactions. The phosphorescent 6-bromo-2-naphthol (BrNp) embedded in the gel exhibited a fluorescent phosphorescent double emission due to the ordered microstructure of the supramolecular gel, thereby effectively avoiding the non-radiative transition of BrNp. Moreover, owing to its excellent biocompatibility and low biotoxicity, HmeQ[5]/DB-BrNp could combine with HeLa cells to achieve cytoplasmic staining in the red channel. This study broadens the application of Q[n]-cross-linked small-molecule gels and reveals their potential applications in novel, phosphorescent, soft materials.

### 3.3. Hydrogels Driven by a Simple Molecule Inclusion Interaction

Supramolecular hydrogels formed by the inclusion interaction between Q[n] and organic small molecules mainly used Q[n] with larger cavities, such as Q[7] and Q[8]. Q[n] has a wider range of binding to guest molecules, which can encapsulate larger guest molecules such as amantadine, ferrocene, and bipyridine.

In 2017, Scherman and co-workers reported the formation of supramolecular hydrogels using simple molecules and Q[8] ternary complexes [43]. As shown in Figure 7, the hydrogel was composed of Q[8] and guest functional molecules (A and B) through host–guest inclusion interactions. After the water-soluble complex was attracted to the liquid–liquid interface via emulsification and electrostatic attraction, the concentration and density of the polymer increased rapidly, with approximately 2 s to cross-link and gel. The addition of guest B with a chiral azobenzene group made the prepared hydrogel responsive to light and chemicals. The hydrogel observed at the micrometer scale was exceptionally gelatinized at the liquid–liquid interface, which could act as a soft matter barrier to inhibit the coalescence of water droplets after the displacement of surfactants.

Similarly, Qu et al. designed an achiral guest 4,4′-bipyridin-1-ium chloride (BPY+) salt derivative and Q[8] to construct a ternary complex by a 2:1 host–guest interaction in an aqueous solution. The flexible chain of the monomer provides moderate water solubility, and the increased solubility in solutions driven by the host–guest interaction between Q[8] and BPY+ groups leads to the spontaneous assembly and formation of helical nanofibers. By varying the guest concentration, the helical nanofibers were transformed into pH-responsive hydrogels, thereby tuning the morphology of the material [22].

## 4. Biomedical Application

As cross-linked three-dimensional network structure materials, conventional hydrogels can absorb and retain a large amount of water [44]. The introduction of noncovalent interactions changes the mechanical strength and self-healing properties of conventional hydrogels [45]. Therefore, supramolecular hydrogels are widely used in biosensors, 3D printing, drug delivery systems, and interfacial adhesion due to their highwater content, softness, biocompatibility, and self-healing. These properties also allow for hydrogels to mimic extracellular matrices (ECM), providing mechanical cues to encapsulated cells, allowing for the study of their impact on cellular behaviors [46].

### 4.1. Cell Encapsulation

Cell encapsulation can prevent biological macromolecules, such as enzymes and cells, escaping from the microcapsules, while small molecule substances and nutrients in the medium can freely egress and ingress, thereby achieving the purpose of catalysis or cultivation. It is an important challenge to develop effective crosslinking and biofunctionalization strategies under physiological conditions without compromising the function of cell encapsulation [47]. The introduction of host–guest interactions enables the hydrogel system to undergo dynamic crosslinking, which can enhance its responsiveness to cellular forces by breaking and reforming crosslinking junctions. Webber et al. reported thermoresponsive supramolecular hydrogels with light-mediated kinetic control, which was appended at two ends of Pluronic F-127 polymers with Q[8]-related guests [48]. In particular, the specific guest which constitutes the ternary complex can be further photoisomerized to form a covalent bond instead of the host–guest linkage, resulting in a decrease in hydrogel dynamics without changing the temperature responsiveness and network architecture. Hydrogels composed of micelles cross-linked via supramolecular and photodimerization interactions can rapidly encapsulate NIH3T3 fibroblasts and be injected into patients without disrupting the integrity of the cells. Similarly, Myung et al. [49] reported the preparation of novel supramolecular hydrogels via thiol-ene reactions between preassembled Q[8] complexes and grafted norbornenes, which extended the encapsulation time to seven days and exhibited a broad distribution in the culture.

### 4.2. Biosensor

Hydrogels are very promising for fabricating transductive biosensors because of their inherent cellular similarities [50]. Supramolecular hydrogels possess rapid signal transmission capabilities due to their rapid phase transitions and reusability due to their reversibility, which lead to high sensitivity and low fabrication costs for biosensor structures. Scherman and co-workers reported a striking, glass-like hydrogel as a pressure sensor in 2022 [51]. Although supramolecular hydrogels are characterized by strong toughness and self-healing ability, it is still a big challenge to fabricate materials that can withstand large pressure without being crushed. Differently from previous soft and stretchable rubber-like hydrogels, this novel, hard, glass-like hydrogel exhibited high compressibility—up to 100 MPa in compressive strength. In addition to the adjustable compressive strength, they can also control the mechanical properties of the material from rubbery to glass-like by changing the molecular structure of the guest molecule. These extremely compressible and ultra-high-strength hydrogel materials are an important milestone for high-performance soft materials. High-performance pressure sensors can be applied in fields such as electronic skin, wearable devices, and soft robots. In order to highlight the application in the field of biosensors, they used hydrogel to prepare a capacitive pressure sensor, whereby the surface was made into a hemispherical structure to increase the sensitivity. This hemispherical structured sensor can withstand ultra-high pressure up to 2.5 MPa and showed 3~4-times higher sensitivity than planar pressure sensors, thus making it potentially applicable in prosthetics, sensing touch of the arms, and robotic skin.

### 4.3. 3D Printing

Hydrogel inks are particularly useful in biomedical applications to reproduce features of the cellular environment, and can form structures used to replace, enhance, and mimic tissue [52]. Hydrogels are similar to the microenvironment of cell growth, are hot materials used for tissue engineering, and can be used as bioinks for 3D printing [53]. The bioinks are extruded into support gels through host–guest interactions to directly write structures continuously in 3D space. The host–guest cross-linked hydrogels are based on modified hyaluronic acid (HA), owing to its amenability to chemical modification and biocompatibility [38]. Kim et al. developed a method for the 3D bioprinting of a multilayered construct consisting of HA-based hydrogels formed through the host–guest interaction between Q[6]/1,6-diaminohexane and pepsin-treated collagen (atelocollagen) with human turbinate-derived mesenchymal stromal cells for the regeneration of osteochondral tissues [54]. The mechanically stable, host–guest chemistry-based hydrogel used two different types of extracellular matrix hydrogels for easy printability and was stacked into one multilayered construct without using potentially harmful chemical reagents and physical stimuli for post-crosslinking. This study validates the potential of 3D-printed multilayer structures composed of two different materials in heterogeneous tissue regeneration using an in vivo animal model. The 3D-printing-based platform technology can be effectively used for regenerating various heterogeneous and osteochondral tissues. For improving the biocompatibility of hydrogel structures, Zheng et al. used an ionic carbazole-based water-soluble femtosecond laser two-photo polymerization photoinitiator to prepare a 3D hydrogel structure in an aqueous phase [55]. The hydrogel is combined with the host–guest inclusion complex interaction between 3,6-Bis[2-(1-methyl-pyridinium)vinyl]-9-methyl-carbazole diiodide (BMVMC) and Q[7], which further improves the solubility, manufacturing ability, and biocompatibility. The 3D engineering hydrogel scaffold microstructure prepared by two-photo polymerization technology has proven its biocompatibility through living cells and is expected to be further applied in tissue engineering.

### 4.4. Drug Release

Owing to its 3D network structure, the hydrogel that consists of hydrophilic polymer chains can encapsulate drug molecules to form potential drug delivery systems [56,57]. The inclusion of drugs in hydrogels to achieve a controlled and sustained drug release is vital for new clinical treatments. Due to the host–guest responsiveness inherited from Q[n], Q[n] has the potential to be an ideal scaffold for drug delivery systems [58,59,60]. This host–guest chemistry has been extensively developed, and a dynamic aggregate formation between Q[n] and coordinated guest-modified polymers has also been extensively demonstrated. Therefore, controllable hydrogels can be prepared by self-assembly and used as drug delivery platforms. There are various natural polymers in nature, such as polysaccharides. A hydrogel with good biocompatibility and biodegradability can be obtained by modifying the existing polysaccharides and crosslinking [61]. Additionally, it has been proven in experiments that the non-covalent supramolecular crosslinking strategy can maintain the biological properties of polysaccharides [57]. In addition, polysaccharide-based particles and hydrogels mediated by host–guest interactions can achieve more specific drug releases through their responses to stimuli, including light, redox, and enzymes. Tan et al. [62] observed nano-sized micelle-like aggregates of sodium alginate upon the addition of Q[6] in an aqueous solution, owing to the exclusion interaction between Q[6] and sodium ions. Moreover, Q[6]/sodium alginate hydrogel beads with uniform size were prepared for the controlled release of the simulated drug 5-fluorouracil [63]. Wang et al. [64] reported the fabrication of chitosan (CS)-based supramolecular nanostructures mediated by the homoternary host–guest interaction of Q[8]. CS was modified with phenylalanine (Phe) through an amidation reaction to obtain CS-Phe. Because a homoternary complex was formed between Q[8] and 2Phe, Q[8] was added to the CS-Phe aqueous solution to form CS nanoparticles (Q[8]/CS-Phe). The model drug doxorubicin was loaded on to Q[8]/CS–Phe nanoparticles, which exhibit a selective drug release after the addition of endogenous and exogenous guests, spermine and amantadine, respectively. In addition, a special interpenetrating double-network hydrogel introduced by Li et al. can be degraded by relevant enzymes to achieve a controllable and responsive drug release [33].

### 4.5. Interfacial Adhesion

In recent years, numerous studies have been conducted on bioadhesives as soft and wet materials used in medicine and engineering, including soft electronics [65], biomedical devices [66], and wound dressings [67]. Owing to their intrinsic properties and functionalization methods, hydrogels have potential as bioadhesives. Compared with covalent bonds, noncovalent bonds can coordinate with each other through multiple reversible interactions. These can stimulate the development of new biomimetic strategies for achieving robust and reversible adhesion at the hydrogel/tissue interface. Scherman et al. proposed that a Q[8]-based supramolecular hydrogel network could be used as a dynamic adhesive to various nonporous and porous substrates [27]. The supramolecular hydrogel network was photoinitiated by a comonomer (1-benzyl-3-vinylimidazolium bromide), which can be complexed with Q[8] in 2:1 of noncovalent supramolecular crosslinkers and hydrophilic comonomer acrylamide/(meth)acrylates [34,68]. We incorporate Q[n]s molecular recognition into adhesives to improve the interfacial toughness between different substrates, and to achieve reversible adhesive and self-healing properties. Owing to their good adhesion ability and excellent self-healing performance, the hydrogel adhesives have potential applications in smart actuators, stretchable soft electronics, hybrid systems, implantable biomedical devices, and tissue bone regeneration.

## 5. Conclusions and Future Prospects

This paper mainly discusses the recent preparation methods of Q[n]-based supramolecular hydrogels and their applications in the field of biomedicine. The unique structure of Q[n] makes it have stable properties, excellent molecular recognition ability, and strong binding ability with guest molecules, which leads to an easy non-covalent bond interaction. As specific stereochemical building blocks in the system, supramolecular hydrogel can be found not only by the self-induced outer-surface interaction, but also by the host–guest inclusion or exclusion interaction. Utilizing the interaction between Q[n] and the guest, combined with the properties of different functional guests, the researchers prepared self-healing, adhesive, stimuli-responsive or biocompatible hydrogels. The low toxicity of Q[n] and the biocompatibility of some hydrogels that can be used as an extracellular matrix make it prone to be applied in biomedical fields, such as cell encapsulation, biosensors, 3D printing, drug releases, and interfacial adhesion. However, the application of supramolecular hydrogels still faces many practical challenges due to the refractory and long-term toxicity of various chemical groups. In order to enrich the application, the development of biosafety, degradable hydrogels, and high mechanical strength is the main focus of current research. With relatively strong biocompatibility of the self-assembled system, supramolecular hydrogels are expected to be a good application for bioscience and medicine. On the other hand, the supramolecular bonding strength is controllable by designing suitable guests, and the regulation of mechanical strength with self-healing in the Q[n]-based hydrogel materials should be realized with supramolecular functionalization. Therefore, the development of the application of Q[n]-cross-linked guest hydrogels is also one of the solutions to the current problems. The Q[n]-based supramolecular hydrogel still has great potential application, providing unlimited possibilities for the research of cucurbit[n]uril chemistry.

## Figures and Tables

**Figure 1 molecules-28-03566-f001:**
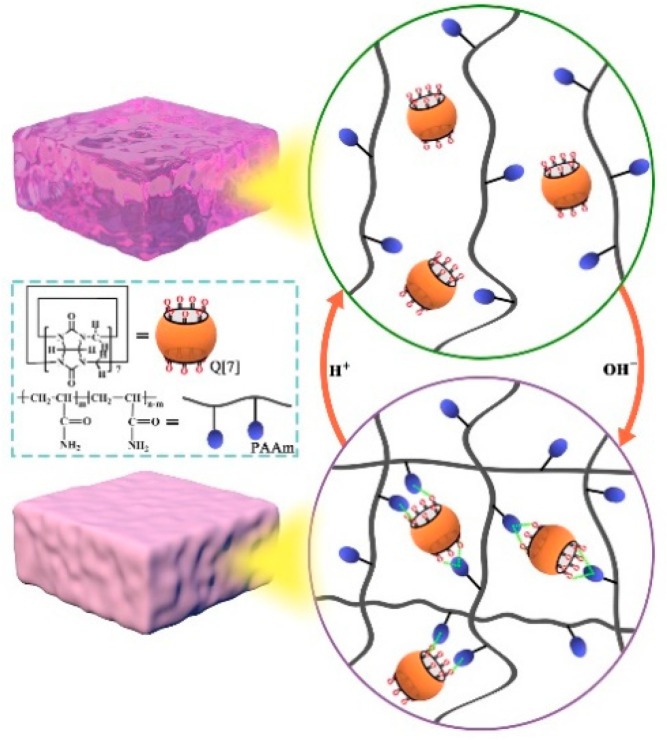
The sol–gel phase transition of Q[7]-PAAm gel.

**Figure 2 molecules-28-03566-f002:**
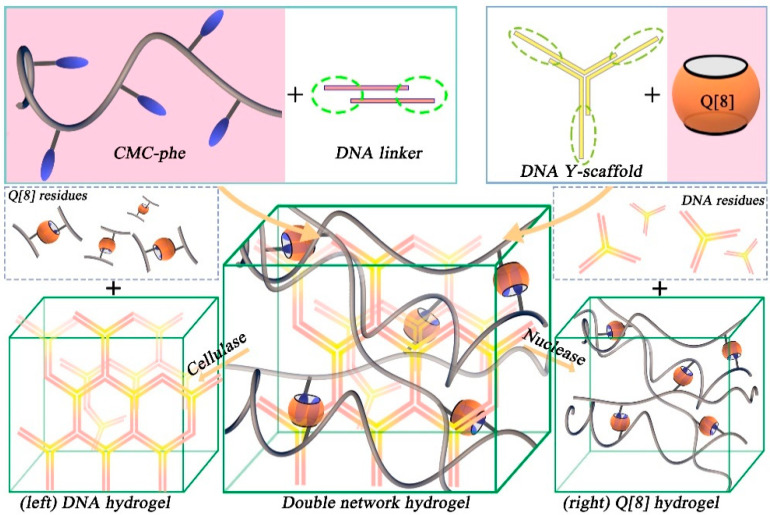
The formation process of double-network hydrogels through supramolecular interactions. The first network is cross-linked by the hybridization of DNA sticky ends (left); the second network is a 2:1 homoternary complexation of a phenylalanine unit pendant from carboxymethyl cellulose to Q[8] (right).

**Figure 3 molecules-28-03566-f003:**
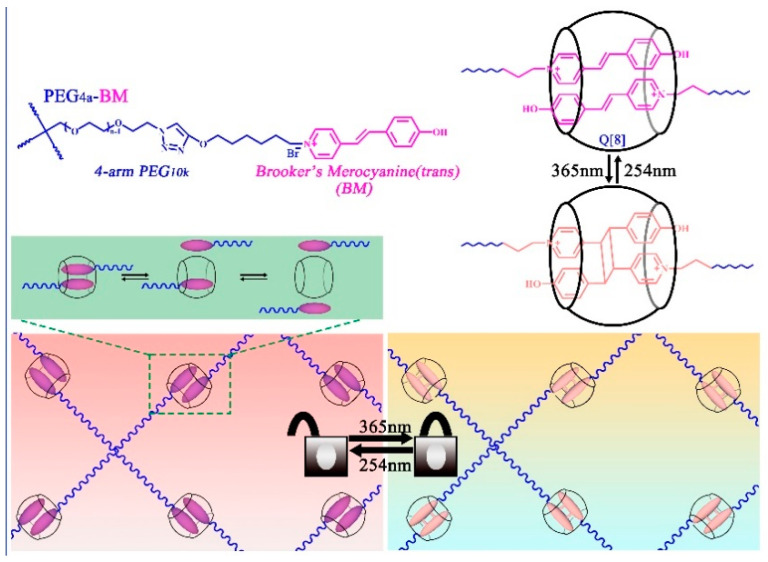
Light-controlled supramolecular hydrogels.

**Figure 4 molecules-28-03566-f004:**
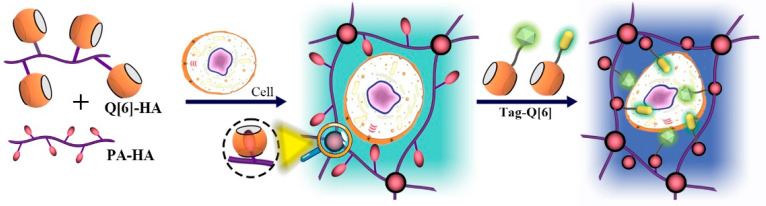
Schematic diagram of the in situ formation of supramolecular biocompatible hydrogels via the host–guest interaction between Q6 and DAH or PA.

**Figure 5 molecules-28-03566-f005:**
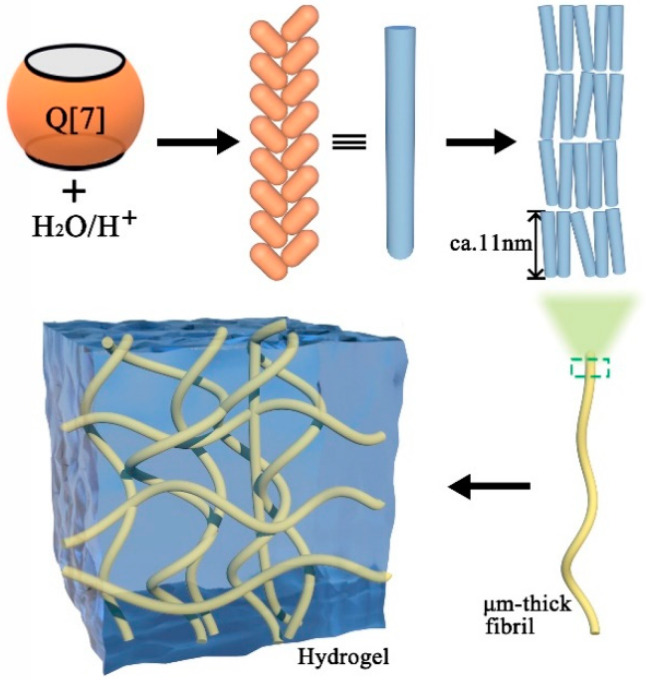
Q[7]-based hydrogels driven by the outer-surface interaction.

**Figure 6 molecules-28-03566-f006:**
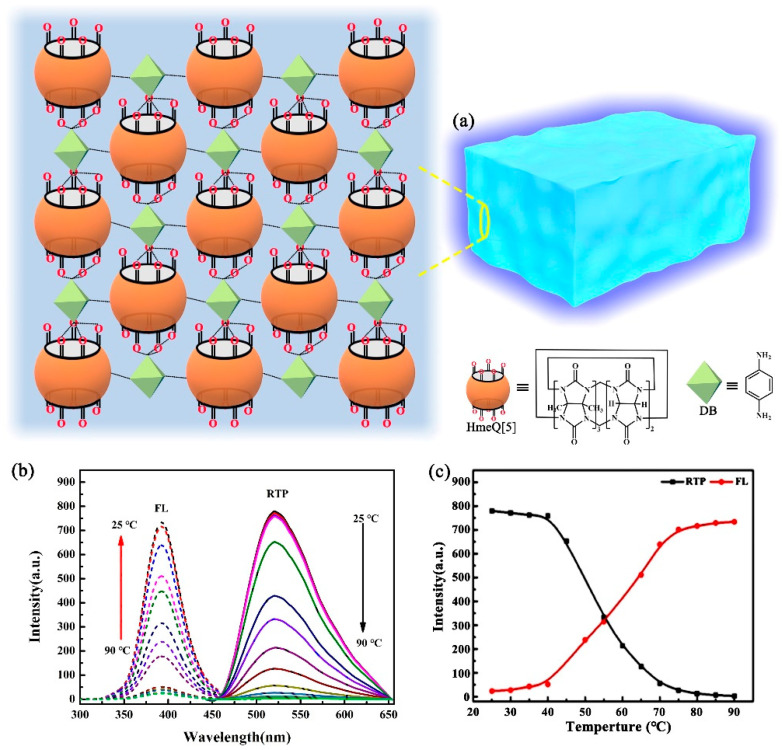
(**a**) HmeQ[5]/p-PDA supramolecular hydrogels by the host–guest exclusion interaction; (**b**) fluorescence and phosphorescence spectra at different temperatures; (**c**) variation in fluorescence and phosphorescence intensity versus temperature.

**Figure 7 molecules-28-03566-f007:**
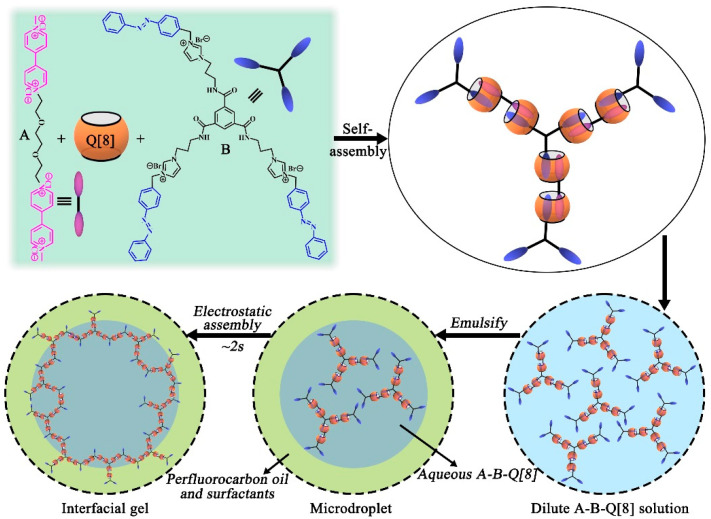
The preparation of the supramolecular hydrogel by Q[8] and double-guest small molecules through the host–guest inclusion interaction.

## Data Availability

Not applicable.

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
