# Peer review of "Preparation and Biomedical Applications of Cucurbit[n]uril-Based Supramolecular Hydrogels"

_molecules, 2023, doi:10.3390/molecules28083566_

Round 1

Reviewer 1 Report

I suggest minor corrections.

1. In Abstract you may give some more details about biomedical applications of hydrogels

2. For Figures, you should add the reference if are not original

3. In References section, you may add the doi code of each article

Author Response

Dear Editors and Reviewers,

Thank you for your letter and the reviewers’ comments concerning our manuscript entitled “Preparation and Biomedical applications of cucurbit[n]uril-based supramolecular hydrogels” (molecules-2333658). The comments raised by reviewers are all valuable and very helpful for improving our manuscript. All matters raised by the reviewer have now been addressed. All revisions made in response to specific referees’ comments are shown in red in the revised version. Our responses to the reviewers’ comments are as following.

Yours faithfully,

Prof. Dr. Hang Cong

Enterprise Technology Center of Guizhou Province

Guizhou University,

Guiyang, 550025

China

  1. In Abstract you may give some more details about biomedical applications of hydrogels.

Thank you for your suggestion. More information on biomedical applications of hydrogels has been included in the revised abstract.

  1. For Figures, you should add the reference if are not original.

Thank you for your comments. All figures in the manuscript were obtained from original publications directly, and promised with copyright transfer.

  1. In References section, you may add the doi code of each article.

Thank you for your suggestion. The doi codes of all references have been included in this revision.

Reviewer 2 Report

The article by Rui-Han Gao et al. discusses the recent preparation methods of Q[n]-based supramolecular hydrogels and summarizes the construction and application of Q[n]-based supramolecular hydrogels. The literature is up to date and provides a good background for the stated objectives. The article is clear, concise, and logical. The article has the following comments:

1. Provide more detailed information on the current development of Q[n]-based supramolecular hydrogels, such as mechanical performance tests, etc.

2. The article lists some advantages of supramolecular hydrogels, such as good reversibility and stimuli response, but poor mechanical properties limit their applications. What are the prospects for improving its mechanical properties?

3. Give specific suggestions for studying the role of Q[n]s in supramolecular hydrogels, as a theoretical and practical basis for subsequent research and practical applications.

4. The novelty of the paper needs to be improved.

The article needs minor revision for language and grammar.

Author Response

Dear Editors and Reviewers,

Thank you for your letter and the reviewers’ comments concerning our manuscript entitled “Preparation and Biomedical applications of cucurbit[n]uril-based supramolecular hydrogels” (molecules-2333658). The comments raised by reviewers are all valuable and very helpful for improving our manuscript. All matters raised by the reviewer have now been addressed. All revisions made in response to specific referees’ comments are shown in red in the revised version. Our responses to the reviewers’ comments are as following.

Yours faithfully,

Prof. Dr. Hang Cong

Enterprise Technology Center of Guizhou Province

Guizhou University,

Guiyang, 550025

China

  1. Provide more detailed information on the current development of Q[n]-based supramolecular hydrogels, such as mechanical performance tests, etc.

Thank you for your suggestion. Since few publications reported mechanical and rheological properties of Q[n]-based supramolecular hydrogels, a full discussion cannot be supported and included in this review.

  1. The article lists some advantages of supramolecular hydrogels, such as good reversibility and stimuli response, but poor mechanical properties limit their applications. What are the prospects for improving its mechanical properties?

Thank your comments. With the pioneering works, the improvement of mechanical properties for Q[n]-based supramolecular hydrogels could include double cross-linking and double network into the construction, and enhancement of host–guest interactions by screening of guest molecules. As mentioned in the response to question 1, no enough examples have been achieved to provide a clear clue for the discussion on the mechanical properties.

  1. Give specific suggestions for studying the role of Q[n]s in supramolecular hydrogels, as a theoretical and practical basis for subsequent research and practical applications.

Thank you for your suggestion. we have added it on p.11of the manuscript.

  1. The novelty of the paper needs to be improved.

Thank your suggestion. More comments have been provided in the revised conclusion.

  1. The article needs minor revision for language and grammar.

Thank your comment. The manuscript has been checked carefully, and few typos and grammatical mistakes have been removed.